# Do Semaphorins Play a Role in Development of Fibrosis in Patients with Nonalcoholic Fatty Liver Disease?

**DOI:** 10.3390/biomedicines10123014

**Published:** 2022-11-23

**Authors:** Lara Šamadan, Neven Papić, Maja Mijić, Ivana Knežević Štromar, Slavko Gašparov, Adriana Vince

**Affiliations:** 1School of Medicine, University of Zagreb, 10000 Zagreb, Croatia; 2Department for Viral Hepatitis, University Hospital for Infectious Diseases Zagreb, 10000 Zagreb, Croatia; 3Department of Internal Medicine, Division of Gastroenterology, University Hospital Merkur, 10000 Zagreb, Croatia; 4Department of Internal Medicine, Division of Gastroenterology and Hepatology, University Hospital Center Zagreb, 10000 Zagreb, Croatia; 5Department of Pathology, University Hospital Merkur, 10000 Zagreb, Croatia

**Keywords:** NAFLD, nonalcoholic fatty liver disease, fibrosis, semaphorins, SEMA3A, SEMA3C, SEMA5A, SEMA7A, steatosis, cirrhosis

## Abstract

Nonalcoholic fatty liver disease (NAFLD) is associated with systemic changes in immune response linked with chronic low-grade inflammation and disease progression. Semaphorins, a large family of biological response modifiers, were recently recognized as one of the key regulators of immune responses, possibly also associated with chronic liver diseases. The aim of this study was to identify semaphorins associated with NAFLD and their relationship with steatosis and fibrosis stages. In this prospective, case-control study, serum semaphorin concentrations (SEMA3A, -3C, -4A, -4D, -5A and -7A) were measured in 95 NAFLD patients and 35 healthy controls. Significantly higher concentrations of SEMA3A, -3C and -4D and lower concentrations of SEAMA5A and -7A were found in NAFLD. While there was no difference according to steatosis grades, SEMA3C and SEMA4D significantly increased and SEMA3A significantly decreased with fibrosis stages and had better accuracy in predicting fibrosis compared to the FIB-4 score. Immunohistochemistry confirmed higher expression of SEMA4D in hepatocytes, endothelial cells and lymphocytes in NAFLD livers. The SEMA5A rs1319222 TT genotype was more frequent in the NAFLD group and was associated with higher liver stiffness measurements. In conclusion, we provide the first evidence of the association of semaphorins with fibrosis in patients with NAFLD.

## 1. Introduction

Nonalcoholic fatty liver disease (NAFLD) is the most common liver disease, with a prevalence of about 25–30% in the Western population [1,2]. Currently, NAFLD is considered a multisystemic disease associated with complex immune changes that include impaired cellular immunity with an imbalance of the Treg/Th17 axis associated with the increased release of proinflammatory cytokines [3,4]. This contributes to the progression of liver inflammation and fibrosis, insulin resistance, cytokine imbalance and chronic low-grade inflammation [5,6]. However, the pathophysiology and drivers of inflammation in NAFLD are largely unknown. 

Semaphorins are one of the key regulators of the immune response, which could play a role in liver diseases as well [7,8]. Semaphorins are a large family of secreted and membrane-bound biological response modifiers that regulate development, angiogenesis, and oncogenic transformation [7,8,9,10,11]. Recently, their role in cellular immunity and inflammatory response have been described; they can stimulate inflammation through the production of inflammatory cytokines or suppress immune cell activation [7,8].

Previously, we have shown that semaphorins play a role in the progression of fibrosis in chronic hepatitis C (CHC) [12]. Serum concentrations of SEMA3C, SEMA5A and SEMA6D correlated with the fibrosis stage and showed better diagnostic accuracy than the AST to Platelet Ratio Index (APRI) and Fibrosis 4 (FIB-4) score in predicting the presence of CHC cirrhosis [12]. Although several gene expression studies suggested that the semaphorin signaling pathway might be enriched in patients with NAFLD [13,14,15], the association of semaphorins with the steatosis and fibrosis stages has not been reported, and their role in NAFLD is unknown.

Based on the growing evidence of their crucial role in regulating immune responses, we hypothesize that semaphorins are regulators of inflammation and novel biomarkers of NAFLD. Therefore, the aim of this study was to identify semaphorins associated with NAFLD and to investigate possible relationship with steatosis and fibrosis stages.

## 2. Materials and Methods

### 2.1. Study Design, Patients and Samples

The prospective study included 95 adult patients diagnosed with NAFLD at the University Hospital for Infectious Diseases Zagreb, University Hospital Centre Zagreb and University Hospital Merkur, in Croatia between 2020 and 2021 (SepsisFAT, ClinicalTrials.gov Identifier: NCT04573543). As healthy controls 35 voluntary blood donors without relevant comorbidities, obesity or history of alcohol abuse, with negative HIV and viral hepatitis markers and normal liver ultrasound and controlled attenuation parameters (CAP) were included.

NAFLD was diagnosed according to current guidelines that require [1,2]: (1) evidence of liver steatosis measured by controlled attenuation parameter (CAP), (2) no significant alcohol consumption, (3) no competing causes of liver steatosis and (4) no coexisting causes of chronic liver disease (including viral hepatitis, which was excluded by testing for HCV antibodies and HBsAg). All patients tested negative for HIV.

Whole blood samples were collected by venipuncture into sterile tubes without anticoagulant and the blood was allowed to clot for 20 min at room temperature. The serum was separated by centrifugation at 2000× *g* for 10 min. The samples were divided into aliquots to avoid repeated freeze/thaw cycles and stored at −80 °C until testing.

Additionally, immunohistochemistry was performed on specimens obtained from 10 patients at the Liver Transplant Center (University Hospital Merkur, Zagreb, Croatia). Three specimens were obtained from healthy liver donors immediately after the explantation.

The study conformed to the ethical guidelines of the Declaration of Helsinki and was approved by the School of Medicine, University of Zagreb Ethics Committee (code 641-1/19-02/01). All participants gave written informed consent.

### 2.2. Laboratory and Clinical Data

The degree of steatosis was estimated using the controlled attenuation parameter (CAP), a method for grading steatosis by measuring the degree of ultrasound attenuation by hepatic fat using a process based on simultaneous transient elastography (TE), which measures the degree of fibrosis [16]. Abdominal ultrasound was performed in all patients to exclude other liver pathologies.

Steatosis was defined as grade I if a patient had CAP > 250 dB/m (S1 ≥ 10% of hepatocytes with fat), grade II CAP > 280 dB/m (S2 ≥ 33% of hepatocytes with fat) and grade III CAP > 300 dB/m (S3 ≥ 66% of hepatocytes with fat) [16,17]. Cutoff values for staging fibrosis were: F1 > 5.5 kPa, F2 > 7.0 kPa, F3 > 9.5 kPa, F4 > 11.5 kPa [16].

Demographic and comorbidity data (including the presence of components of metabolic syndrome, cardiovascular, kidney and neurological conditions), chronic medications and baseline clinical status were collected in a standardized form. Anthropometric measures, including body mass index (BMI), waist circumference (WC), waist-hip ratio (WHR) and waist-height ratio (WHtR) were measured in all patients.

Results of the routine laboratory tests as part of the standard diagnostic procedure were collected: bilirubin, aspartate aminotransferase (AST), alanine aminotransferase (ALT), gamma-glutamyl transferase (GGT), alkaline phosphatase (ALP), albumins, white blood cell count (WBC), neutrophil-to-lymphocyte ratio, hemoglobin, platelet count, fasting glucose, triglycerides, cholesterol, high-density lipoprotein (HDL) and low-density lipoprotein (LDL). Non-invasive scores of steatosis/fibrosis were calculated (APRI, FIB4, AST/ALT ratio) [18,19].

### 2.3. Semaphorin Measurements

The quantification of semaphorins was performed by using the standardized ELISA assays (Human Semaphorin -3A, -4A, -4D, -5A, and -7A by ELISA kit, AssayGenie, Dublin, Ireland, and Human Semaphorin -3C ELISA Kit, MyBioSource, San Diego, CA, USA), as recommended by the manufacturer. Quantification ranges of semaphorin ELISAs were as follows: SEMA3A (0.156–10 ng/mL), SEMA4A (0.312–20 ng/mL), SEMA4D (0.312–20 ng/mL), SEMA3C (0–10 ng/mL), SEMA5A (0.312–20 ng/mL) and SEMA7A (0.156–10 ng/mL).

### 2.4. Immunohistochemical Staining

Tissue blocks from each patient diagnosed with NAFLD were cut to 2 μm thick slides and prepared for immunohistochemical staining for detection of SEMA4D, SEMA5A and SEMA7A proteins. After tissue slides deparaffinization, heat induced epitope retrieval (HIER) was used prior to detection of specific proteins with polymer-based detection systems EnVision (Dako/Agilent, Santa Clara, CA, USA) according to the manufacturer’s instructions. All staining was performed by automated immunostainer (Autostainer Link 48, Dako/Agilent, Santa Clara, CA, USA). Primary antibodies used were rabbit monoclonal anti-semaphorin 4D/CD100 antibody (1:100), goat polyclonal anti-semaphorin 5A (1:125) and rabbit monoclonal SEMA7A (1:100) (Abcam, Cambridge, UK). Human tonsils were used as a positive control for SEMA4D and SEMA7A and human kidney tissue as a positive control for SEMA5A. For the negative control, adjacent tissue sections were stained without the primary antibody. Samples were analyzed, by two independent researchers using Olympus BX51 microscope (Olympus Corporation, Shinjuku City, Tokyo, Japan). 

### 2.5. Semaphorin SNP Analysis

DNA was extracted from the whole blood samples using the QIAGEN QIAamp DNA Blood Mini Kit spin method, according to the manufacturer′s instructions. The DNA concentration was determined using the CLARIOstar Plus. The genotypes were determined by polymerase chain reaction (PCR) using commercially available TaqMan SNP assays (Thermo Fisher Scientific, Life Sciences Solutions Group, Waltham, Massachusetts USA) for SEMA5A (rs1319222 and rs433755), SEMA4A (rs536857) and SEMA3A (rs17158675 and rs76881522). These were selected based on their possible associations with metabolic syndrome and NAFLD available in NHGRI-EBI GWAS Catalog [20].

### 2.6. Statistical Analysis

The clinical characteristics, laboratory and demographic data were evaluated and descriptively presented as frequencies and medians with interquartile ranges. Comparative statistics between two or more groups were generated using the chi-square test, Mann–Whitney U or the Kruskal–Wallis test, as appropriate. All tests were two-tailed; a *p*-value < 0.05 was considered statistically significant. Correlations were analyzed using Spearman’s rank correlation coefficient and summarized in a correlation matrix. The discriminatory performances of the laboratory variables considered were compared using a receiver operating characteristic (ROC) analysis. Statistical analyses were performed using GraphPad Prism Software version 9.4.1 (San Diego, CA, USA).

## 3. Results

### 3.1. Baseline Patients’ Characteristics

A total of 95 patients with NAFLD (51.6% males; median age of 54, IQR 44–56 years) and 35 controls (40% males; median age of 41, IQR 31–52 years) were included. Patients with NAFLD more frequently had diabetes mellitus (30.5% vs. 8.6%), arterial hypertension (49.5% vs. 8.6%), dyslipidemia (28.4% vs. 11.4%) and obesity (BMI 31 kg/m^2^ vs. 24 kg/m^2^). The median CAP in patients with NAFLD was 307 (IQR 283–343 db/m) and in healthy controls 206 (IQR 191–230). The majority of NAFLD patients had grade 3 steatosis (58, 61%), 22 (23%) had grade 1 and 15 (16%) had grade 2; 39 patients (41%) with NAFLD had no fibrosis (F0), and in 27 (28%), 15 (16%), 6 (6%) and 8 (8%) patients, F1 to F4 scores, respectively, were detected. The characteristics of NAFLD patients according to fibrosis stage are presented in Table 1.

### 3.2. Serum Concentrations of Semaphorins in Patients with NAFLD

We examined serum semaphorin concentrations in the NAFLD group compared to healthy controls. In comparison to healthy controls, serum levels of SEMA3A, SEMA3C and SEMA4D were significantly higher, whereas SEMA5A and SEMA7A were significantly lower in NAFLD patients, as shown in Figure 1 and in Appendix A. Only SEMA4A was not significantly altered. There was no difference in semaphorin expression in healthy controls regarding sex and age. 

### 3.3. Correlation of Semaphorins’ Concentrations with Steatosis and Fibrosis Stages

Next, we analyzed serum concentrations of semaphorins in different steatosis grades and fibrosis stages. While there was no difference in semaphorin concentrations according to steatosis grades (Figure 1, Appendix A), SEMA3C and SEMA4D significantly increased and SEMA3A significantly decreased with fibrosis stages. Although there was a trend towards lower SEMA5A concentrations in F3/4 fibrosis, it did not reach statistical significance. SEMA7A showed no correlation with the fibrosis stage (Figure 1, Appendix A).

### 3.4. Correlation Analysis of Serum Semaphorin Concentrations with Routine Clinical and Laboratory Findings

We then analyzed potential correlations among paired laboratory parameters, including semaphorin concentrations and clinical variables in patients with NAFLD, as presented in Figure 2. Serum SEMA3A, -3C and -4D positively correlated with CAP, while SEMA5A and -7A showed a negative correlation. Similarly, SEMA3C and -4D showed a positive correlation with BMI, WHR and WHtR as markers of abdominal obesity. SEMA3C positively and -7A negatively correlated with cholesterol, while SEMA4A showed a positive correlation and -4D showed a negative correlation with triglycerides. 

### 3.5. Diagnostic Value of Semaphorins to Detect Steatosis and Progressed Fibrosis

To determine the predictive discriminating value of biomarkers for the detection of steatosis and progressed fibrosis, we performed ROC analyses.

As shown in Figure 3a, all tested semaphorins showed good accuracy in predicting steatosis. A cutoff value of 18.5 ng/mL of the SEMA3A assay correctly predicted NAFLD with a sensitivity of 70% and specificity of 65% (AUC 0.70, CI 95% 0.62–0.78). SEMA3C above 13.5 ng/mL showed a sensitivity of 74% and specificity of 60% (AUC 0.69, 95%CI 0.57–0.82), SEMA4D > 55.2 ng/mL sensitivity of 75% and specificity of 60% (AUC 0.69, 95%CI 0.59–0.79). SEMA5A < 15 ng/mL (sensitivity 65%, specificity 52%, AUC 0.64, 95%CI 0.56–0.73) and SEMA7A < 1.1 ng/mL (sensitivity 65%, specificity 63%, AUC 0.62, 95%CI 0.59–0.71) had lower discriminatory values. 

Secondly, the cutoff values of semaphorin concentrations to correctly predict progressed fibrosis/cirrhosis (≥F3) were calculated, Figure 3b. Compared to the widely recommended FIB-4 score (>1.3, sensitivity 69%, specificity 67%, AUC 0.68, 95%CI 53–83%), SEMA3A, SEMA3C and SEMA4D showed better accuracy levels in our cohort; SEMA3A <17 ng/mL (sensitivity 72%, specificity 70%, AUC 0.74, 95%CI 0.66–0.83), SEMA3C >25 ng/mL (sensitivity 78%, specificity 72%, AUC 0.71, 95%CI 0.77–0.95), SEMA4D > 65 ng/mL (sensitivity 65%, specificity 61%, AUC 0.71, 95%CI 0.61–0.79).

### 3.6. SEMA4D Expression in Liver Tissue Samples

To gain further insight into tissue expression of SEMA4D, SEMA5A and SEMA7A, we performed a series of immunohistochemical staining in NAFLD livers tissues with and without significant fibrosis in comparison to healthy donors. In healthy livers SEMA4D was detected mainly in lymphocytes (Figure 4a). SEMA5A and SEMA7A expression was not detected by immunohistochemistry, neither in hepatocytes, nor in endothelial cells in healthy livers. By contrast, in NAFLD livers, SEMA4D expression was detected in hepatocytes and lymphocytes as well in endothelial cells (Figure 4b,c), and the expression was markedly higher in patients with advanced fibrosis/cirrhosis (Figure 4d). A weak SEMA5A and SEMA7A expression in NAFLD livers was found only in endothelial cells. 

### 3.7. The Frequency of SNP SEMA5A rs1319222, rs433755 and SEMA4A rs536857 in Patients with NAFLD

Next, we investigated the genotype frequency and allele distribution of selected SEMA3A, SEMA4A and SEMA5A SNPs and the possible association between the SNPs, NAFLD and liver fibrosis.

The frequencies of the SEMA5A rs1319222 genotypes TT, GT and GG in the healthy control group were 18.5%, 51.4% and 14.3%, and those in NAFLD patients were 52.2%, 36.2%, and 5.3%, respectively, showing a significantly higher frequency of the TT allele in the NAFLD group (*p* = 0.0031).

There were no significant differences in SEMA5A rs433755 and SEMA4A rs536857 allele distributions, as presented in Table 2. Meanwhile, in all tested samples, the SEMA3A rs17158675 and rs76881522 alleles were undetermined for A/G and T/A, respectively.

### 3.8. The Association of SEMA5A rs1319222 Genotype with Clinical and Laboratory Features in Patients with NAFLD

To further establish the role of the studied SNPs in the development of *NALFD*, we analyzed their association with clinical and laboratory parameters. 

The patients with the SEMA5A rs1319222 GG genotype had significantly higher SEMA5A serum concentrations, followed by the GT genotype, while the TT genotype had the lowest SEMA5A serum concentrations, as presented in Figure 5a. There were no differences in age, sex, presence of obesity, DM, BMI and other laboratory parameters, except for AST levels, which were highest in the TT and lowest in the GT genotype (35 IU/L, 25–44 vs. 26 IU/L, 22–35, *p* = 0.0251). However, we detected significant differences in liver stiffness measurements, Figure 5b. Patients with genotype TT had significantly higher CAP and liver stiffness measurements than patients with the GG genotype. 

Similarly, the SEMA5A rs433755 AA genotype was associated with the lowest and CC genotype with the highest serum SEMA5A concentrations but without difference in other laboratory and clinical findings, Figure 5c,d.

None of the parameters differed between SEMA4A rs536857 genotypes.

## 4. Discussion

It is widely accepted that in NAFLD a persistent low-grade inflammation with sustained cytokine production plays a critical role in the development of liver inflammation, fibrosis, cirrhosis and extrahepatic complications [3,4,21]. However, the immune networks behind NAFLD pathogenesis have not been fully explained. The studies reporting the association of inflammatory mediators and cytokines with NAFLD are inconsistent and often contradictory; while some found positive associations, others reported none [21]. There is growing evidence that the semaphorin signaling pathways influence the outcome of immune responses [7,8]. Here we provide the first evidence that semaphorins might play a role in NAFLD as well.

First, we found increased serum concentrations of class 3 semaphorins, SEMA3A and SEMA3C, in patients with NAFLD. While there was no correlation with steatosis grade, SEMA3C increased and SEMA3A decreased with the fibrosis stage. Furthermore, both SEMA3A and SEMA3C were better predictors of advanced fibrosis than the widely accepted FIB-4 score, at least in our cohort. There are several possible biological explanations. 

It was reported that SEMA3A inhibits neutrophil migration, promotes a transition of classically activated (M1) macrophages toward a resolution phenotype and negatively regulates T-cell-mediated immune responses [22,23,24]. In patients with systemic lupus erythematosus reduced expression of SEMA3A correlated with inflammation and disease severity [25], and exogenous administration of SEMA3A was associated with markedly reduced articular inflammation in the rheumatoid arthritis mouse model [26]. In metabolic syndrome, SEMA3A was shown to have an inhibitory role in adipogenesis [27]. Recent studies also found that SEMA3A regulates the behavior of cancer cells and is highly expressed in HCC, which correlates with metastatic potential and cancer aggressivity [28].

SEMA3C regulates extracellular matrix composition promoting the insulin-resistant state in adipose tissue, including increased expression of IL-6, TGF-ß and CTGF [29]. Moreover, SEMA3C expression correlated with body weight, insulin resistance and adipose tissue morphology [29]. Liver and serum SEMA3C expressions correlate with liver fibrosis in CHC [12]. Recently, De Angelis Rigotti F et al. suggested the possible role of SEMA3C in fibrosis by activating hepatic stellate cells and promoting fibrinogenesis through TGF-β-induced gene expression and SMAD2 phosphorylation [30]. Deletion of either SEMA3C or its receptor NRP2 in activated HSCs reduced liver fibrosis in mice [30].

Therefore, in NAFLD, class 3 semaphorins might have a dual role; inhibition of SEMA3A might promote adipogenesis, insulin resistance, liver neutrophil migration and macrophage- and T-cell-driven inflammation, while increased SEMA3C promotes extracellular matrix composition through TGF-β activation of HSC.

Second, we examined the expression of class 4 semaphorins; SEMA4A stimulates T-cell activation and SEMA4D has crucial stimulatory functions in a broad range of cells [31]. While SEMA4A was not significantly altered (and this was not associated with rs536857 SNP), SEMA4D expression was significantly higher in both serum and NAFLD livers, and its expression correlated with the fibrosis stage. Secreted SEMA4D was shown to enhance neutrophil-mediated inflammatory responses [32]. Several studies reported its role in autoimmunity, as a potent activator of T-cell responses contributes to inflammation by inducing imbalance in Treg/Th17 axis [33]. Specifically in NAFLD, an imbalance of the Treg/Th17 axis and an increased release of proinflammatory cytokines and suppressed negative regulators of the Th17 response are thought to be key inflammatory events [3,4].

Third, we found a lower expression of SEMA5A in patients with NAFLD, regardless of the steatosis and fibrosis stage. SEMA5A was shown to promote T-cell and NK-cell proliferation and induce the secretion of Th17 cytokines that are implicated in RA disease activity, although the functional role has yet not been elucidated [34]. Similarly, we reported a negative correlation with the fibrosis stage in CHC [12]. It has been shown that SEMA5A has a possible role in increased endothelial cell proliferation, migration and decreased apoptosis [35].

Further, we examined the frequency of preselected SNPs of SEMA5A. SEMA5A rs1319222 was selected due to the suggested link with NAFLD severity in the GWAS study [36]. We found a significant association of rs1319222 TT genotype with NAFLD, lower SEMA5A serum concentrations and increased AST, CAP and liver stiffness measurements. To the best of our knowledge, this is the first study describing the association of this SNP with NAFLD. The reference allele at SEMA5A rs1319222 is T, with an alternative G. Such a change is predicted to be benign and does not affect the protein product since it is in the non-coding part of the DNA. According to the GnomAd database, the frequency of this change is 33.15% in the general population [37]. This highlights the possible role of SEMA5A in liver diseases, the mechanism of which is still unknown and should be confirmed in further studies.

Finally, we found a lower expression of SEMA7A in NAFLD. SEMA7A exerts opposite effects by interacting with different receptors and organ systems. While it promotes the secretion of proinflammatory cytokines through the plexin C1 receptor, by the interaction with integrin receptors it has a protective and anti-inflammatory role [38]. Recently it was shown that SEMA7A is crucial for inflammation resolution by promoting polarization of activated macrophages toward the pro-resolving M2 phenotype, leukocyte clearance, and generation of specialized pro-resolving lipid mediators [39]. Interestingly, it was demonstrated that the SEMA7A R148W mutation is a potentially new genetic determinant of NAFLD that promotes intrahepatic lipid accumulation in mice [40].

Our study should be viewed within its limitations. We included a relatively small number of patients; liver fibrosis and steatosis were assessed by non-invasive techniques, not by liver biopsy. Gene polymorphism analysis was performed only for several selected SNPs that should be confirmed in larger population studies. The semaphorin concentrations were measured at a single timepoint, and dynamic changes and correlations with other cytokines were not analyzed. The liver expression levels of semaphorins were determined by IF, and these results should be confirmed by mRNA and Western blot analysis. Nevertheless, we provide the first evidence of the role of semaphorins in NAFLD. This was the first phase of the study aiming to depict the role of semaphorins in NAFLD and infections (project SepsisFAT). While we showed the association of serum concentrations with NAFLD, the question of their pathophysiological role in NAFLD remains unknown. The next step would be to show how they regulate T-cell immunological homeostasis and correlation with cytokine and growth factor networks in NAFLD patients, cell cultures and mice models.

## 5. Conclusions

In conclusion, we demonstrated that circulating levels of SEMA3A, SEMA3C, SEMA4D, SEMA5A and SEMA7A are altered in the serum of patients with NAFLD, and SEMA3A, -3C and -4D concentrations correlated with the stage of fibrosis. Considering their important role in the regulation of immune responses and possible therapeutic potential already being examined in autoimmune and neoplastic diseases, semaphorins should also be further investigated as a new biomarker and therapeutic target in NAFLD.

## Figures and Tables

**Figure 1 biomedicines-10-03014-f001:**
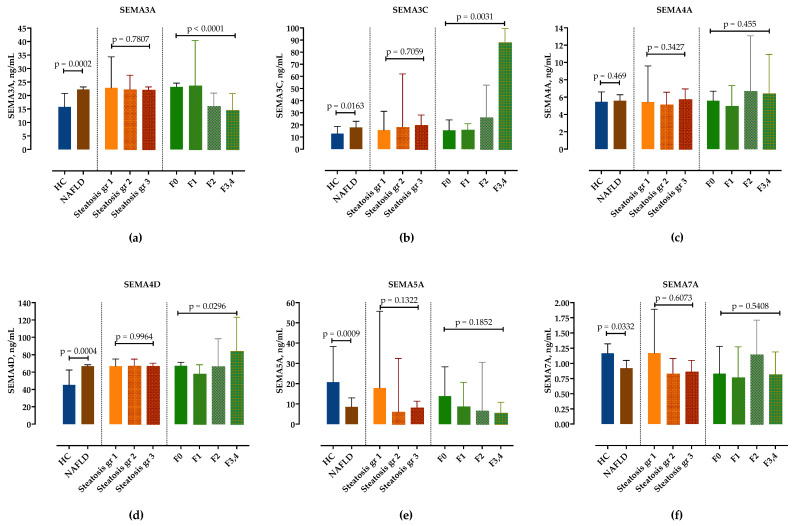
Serum concentrations of semaphorin SEMA3A (**a**), SEMA3C (**b**), SEMA4A (**c**), SEMA4D (**d**), SEMA5A (**e**) and SEMA7A (**f**) measured by ELISA in healthy controls and in patients with NAFLD, stratified by steatosis grade and fibrosis stage. Data are presented as median with interquartile range. The *p*-values are calculated by Mann–Whitney U-test or Kruskal–Wallis test, as appropriate.

**Figure 2 biomedicines-10-03014-f002:**
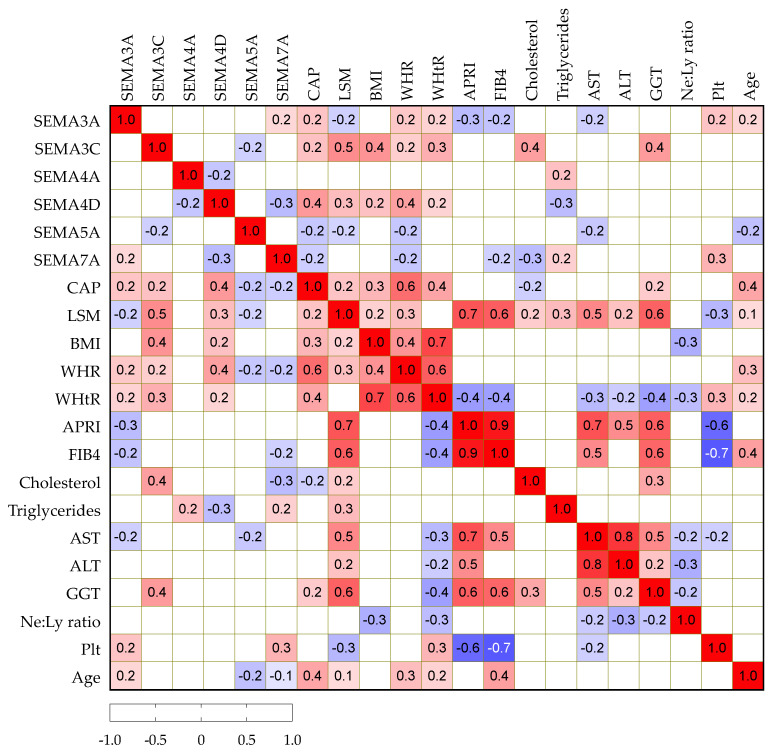
Spearman correlation correlogram. The strength of the correlation between two variables is represented by the color at the intersection of those variables. Colors range from dark blue (strong negative correlation; r = −1.0) to red (strong positive correlation; r = 1.0). Results were not displayed if *p* > 0.05.

**Figure 3 biomedicines-10-03014-f003:**
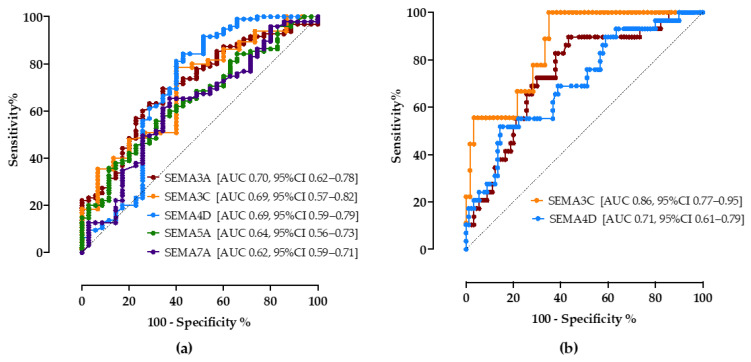
The ROC curve analysis of serum semaphorin concentrations for discrimination of NAFLD (**a**) and progressed fibrosis (**b**). Shown are AUC, area under the curve, with corresponding 95% confidence intervals.

**Figure 4 biomedicines-10-03014-f004:**
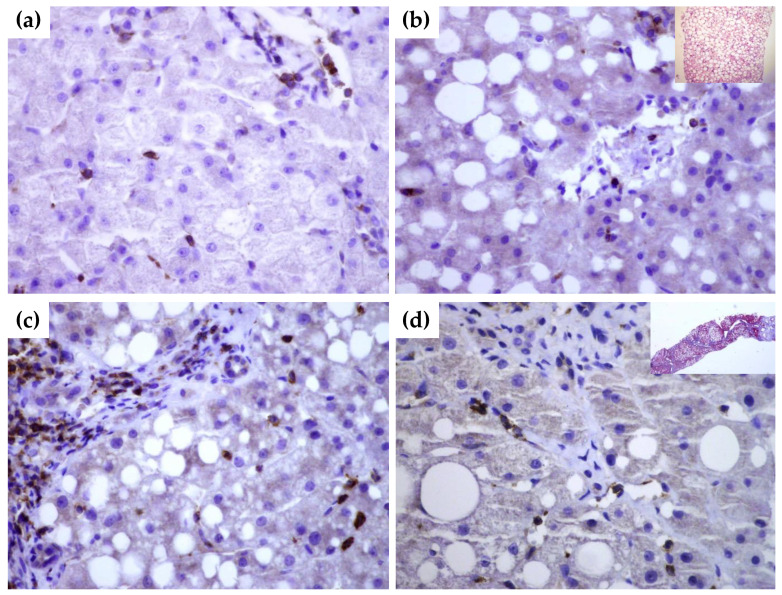
Immunohistochemical staining of SEMA4D in healthy (**a**) and NAFLD livers (**b**–**d**). While there is no expression of SEMA4D marker in hepatocytes of healthy liver tissue ((**a**), magnification 60×), in NAFLD liver without significant fibrosis ((**b**), magnification 60×; in upper right corner hematoxylin-eosin stain showing fat accumulation), there is a moderate expression in hepatocytes, endothelial cells and lymphocytes. In cirrhotic liver ((**c**,**d**), magnification 60×), SEMA4D showed a stronger signal. In the upper right quadrant in panel (**b**), Masson’s trichrome staining of cirrhotic liver is shown.

**Figure 5 biomedicines-10-03014-f005:**
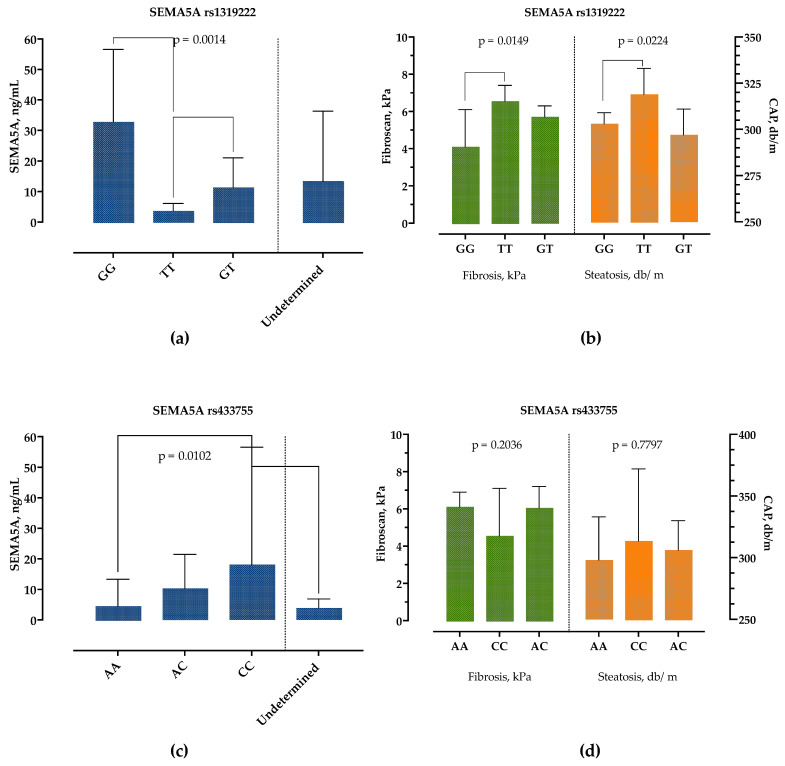
Association of SEMA5A rs1319222 and rs433755 genotype with SEMA5A serum concentrations (**a**,**c**) and liver stiffness (kPa) and controlled-attenuation parameter (db/m) measurements (**b**,**d**). Data are presented as median with interquartile range. The *p*-values are calculated by Kruskal–Wallis test, and significant differences calculated by Dunn’s multiple comparisons test.

**Table 1 biomedicines-10-03014-t001:** Baseline patient characteristics according to steatosis and fibrosis stage.

	No Fibrosis *n* = 39	Fibrosis Stage 1,2 *n* = 42	Fibrosis Stage 3,4 *n* = 14
Age, years	53 (46–65)	55 (42–64)	53 (49–68)
Male sex	12 (30.7%)	29 (69%)	8 (57.1%)
Diabetes mellitus	13 (33.3%)	8 (19.0%)	4 (28.5%)
Hypertension	20 (51.8%)	19 (45.2%)	8 (57.1%)
Dyslipidemia	10 (25.6%)	12 (28.6%)	5 (35.7%)
BMI, kg/m^2^	30 (28–33)	31 (28–33)	32 (27–35)
Waist hip ratio (WHR)	0.91 (0.88–0.98)	0.97 (0.94–0.99)	0.98 (0.88–1.1)
Waist height ratio (WHtR)	0.59 (0.56–0.62)	0.62 (0.58–0.67)	0.57 (0.52–0.63)
Neutrophils/lymphocytes ratio	1.8 (1.5–2.2)	2.0 (1.4–2.6)	1.4 (1.1–2.3)
Bilirubin, µmol/L	11 (8–14)	13 (9–15)	15 (8.5–15)
AST, IU/L	27 (20–34)	30 (24–39)	46 (25–77)
ALT, IU/L	35 (22–56)	44 (32–59)	58 (39–115)
GGT, IU/L	33 (20–77)	51 (30–101)	78 (37–127)
Cholesterol, mmol/L	5.3 (4.2–6.1)	5.3 (4.7–5.9)	6.0 (4.5–6.9)
Triglycerides, mmol/L	1.6 (1.3–2.3)	1.5 (1.1–2.1)	1.7 (1.6–3.3)
LDL, mmol/L	2.9 (2.4–3.8)	3.3 (2.9–4.3)	3.5 (1.9–4.5)
HDL, mmol/L	1.3 (1.0–1.4)	1.2 (1.0–1.6)	1.2 (0.94–1.3)
APRI score	0.28 (0.17–0.35)	0.34 (0.22–0.46)	0.68 (0.27–1.3)
FIB-4 score	1.0 (0.79–1.4)	0.95 (0.66–1.4)	1.4 (0.94–4.0)
Liver stiffness, kPa	4.3 (3.7–4.8)	6.8 (6.1–7.4)	15 (9.9–25)
CAP, dB/m	305 (279–353)	307 (287–335)	330 (290–391)
Steatosis grade 1	11 (28.2%)	9 (21.4%)	2 (14.3%)
Steatosis grade 2	6 (15.4%)	7 (16.7%)	2 (14.3%)
Steatosis grade 3	22 (56.4%)	26 (61.9%)	10 (71.4%)

Presented are baseline patient clinical, laboratory and virological characteristics according to fibrosis stage as measured by liver elastography. Data are presented as frequencies (%) or medians with interquartile ranges.

**Table 2 biomedicines-10-03014-t002:** Distribution of genotypes and allele frequencies in SEMA5A-rs1319222, -rs433755 and SEMA4A-rs536857 in patients with NAFLD and in healthy controls.

	NAFLD *n* = 95	Healthy Controls *n* = 35	*p*-Value ^1^
SEMA5A rs1319222			
TT	50 (52.2%)	6 (18.5%)	
GG	5 (5.3%)	5 (14.3%)	
GT	34 (36.2%)	18 (51.4%)	0.0031
Undetermined	5 (6.3%)	5 (14.3%)	
SEMA5A rs433755			
AA	26 (27.5%)	12 (33.3%)	
AC	41(43.5%)	17 (48.1%)	0.3248
CC	21 (21.7%)	6 (18.5%)	
Undetermined	7 (7.2%)	0	
SEMA4A rs536857			
AA	15 (15.9%)	3 (7.4%)	
AG	25 (26.1%)	10 (29.6%)	0.1089
GG	28 (29.0%)	17 (48.1%)	
Undetermined	28 (29.0%)	5 (14.8%)	

^1^ Chi-square test.

## Data Availability

The datasets generated during and/or analyzed during the current study are available from the corresponding author on reasonable request.

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
