# Peer review of "Do Semaphorins Play a Role in Development of Fibrosis in Patients with Nonalcoholic Fatty Liver Disease?"

_biomedicines, 2022, doi:10.3390/biomedicines10123014_

Round 1

Reviewer 1 Report

The authors investigated serum levels, tissue expression, and genotypes of different semaphorins in non-alcoholic fatty liver disease. The idea is novel and it is a multicenter, registered, prospective study that is well-designed. The results are presented appropriately and they make sense. The discussion is also pertinent and truly relates relevant evidence to their findings. I especially liked their discussion on SEMA5A rs1319222. Overall, I think the manuscript is a piece of valuable knowledge.

Some minor points:

1.     Please include the full form of APRI in the introduction.

2.     Please name the provider/manufacturer of the TaqMan SNP assay used.

3.     Supplementary Table 1: “STEATOSIS GRADE 3” at the left upper corner looks incorrect.

4.     Figure 2: The scale bar below the figure is colorless. I think it should show a gradient from dark blue to dark red.

5.     Figure 3, panel A: It looks that the ROC curve is to predict NAFLD, and not steatosis.

6.     Figure 4. I am afraid that images c and d do not show stronger staining in hepatocytes compared to image b. Please replace them with more illuminating representative images, if available. 

Author Response

The authors investigated serum levels, tissue expression, and genotypes of different semaphorins in non-alcoholic fatty liver disease. The idea is novel and it is a multicenter, registered, prospective study that is well-designed. The results are presented appropriately and they make sense. The discussion is also pertinent and truly relates relevant evidence to their findings. I especially liked their discussion on SEMA5A rs1319222. Overall, I think the manuscript is a piece of valuable knowledge.

Some minor points:            

  1. Please include the full form of APRI in the introduction.

ANSWER: We added the full form of APRI and FIB-4 score. In addition, we added references for FIB-4 and APRI in Methods section.

  1. Please name the provider/manufacturer of the TaqMan SNP assay used.

ANSWER: We added the name of manufacturer (Thermo Fisher Scientific, Life Sciences Solutions Group, USA)

  1. Supplementary Table 1: “STEATOSIS GRADE 3” at the left upper corner looks incorrect.

ANSWER: We thank the reviewer for noticing that error. It is now corrected. Title in Supp. Table 1 is NAFLD.

  1. Figure 2: The scale bar below the figure is colorless. I think it should show a gradient from dark blue to dark red.

ANSWER: We increased the resolution of the graphic.

  1. Figure 3, panel A: It looks that the ROC curve is to predict NAFLD, and not steatosis.

ANSWER: We have corrected Figure 3 legend to NAFLD.  

  1. Figure 4. I am afraid that images c and d do not show stronger staining in hepatocytes compared to image b. Please replace them with more illuminating representative images, if available. 

ANSWER: We thank the reviewer for suggestion. We replaced figure b with more representative image.

Reviewer 2 Report

In this manuscript, Vince et al provided the fisrt evidence on association of semaphorins with fibrosis in patients with NAFLD. But some issues need to be addressed.

1,  Do the patients have other kinds of diseases? and Do the healthy control have some diseases?

2, Figure 4, only SEMA4D in liver sample was stained. The marker of NAFLD and fibrosis should also be stained.

3, authors only processed staining of liver samples. Western blot or RNA assay is also needed to determine the expression level.

4, Have authors determine the results in some cell lines or mouse model?

5, authors stated limitations of this study. Your future plan or perspective should be included.

6, the references in this manuscript were almost published 10 years ago. 

Author Response

In this manuscript, Vince et al provided the fisrt evidence on association of semaphorins with fibrosis in patients with NAFLD. But some issues need to be addressed.

1. Do the patients have other kinds of diseases? and Do the healthy control have some diseases?

ANSWER: The patients’ comorbidities are described in Table 1. Exclusion criteria were immunosuppression, pregnancy, active malignant disease, and no competing causes of liver steatosis and no coexisting causes of chronic liver disease.  Regarding healthy controls, 3 of them had T2DM, 3 arterial hypertension and 3 hyperlipidemia, as stated in Results: “Patients with NAFLD more frequently had diabetes mellitus (30.5% vs 8.6%), arterial hypertension (49.5% vs 8.6%), dyslipidemia (28.4% vs 11.4%) and obesity (BMI 31 kg/m2 vs 24 kg/m2).”

2. Figure 4, only SEMA4D in liver sample was stained. The marker of NAFLD and fibrosis should also be stained.

ANSWER: The liver was also stained by Masson's trichrome stain to visualize liver fibrosis. This is shown in figure 4d, upper right quadrant. This is now described in detail in Figure legend.

“Immunohistochemical staining of SEMA4D in healthy (a) and NAFLD livers (b, c, d). While there is no expression of SEMA4D marker in hepatocytes of healthy liver tissue (a, magnification 60x), in NAFLD liver without significant fibrosis (b, magnification 60x; in upper right corner hematoxylin-eosin stain showing fat accumulation), there is a moderate expression in hepatocytes, endothelial cells and lymphocytes. In cirrhotic liver (c and d, magnification 60x) the SEMA4D showed a stronger signal. In the upper right quadrant in panel b, Masson's trichrome staining of cirrhotic liver is shown.”

3. authors only processed staining of liver samples. Western blot or RNA assay is also needed to determine the expression level.

ANSWER: We agree with the reviewer that WB and gene expression would confirm the data. Unfortunately, we were not able to perform these experiments and they were out of the scope of this manuscript, but they should be performed in future studies. However, we believe that immunohistochemical analysis confirmed serum concentrations.  We have added this in study limitations.

“The liver expression levels of semaphorins were determined by IF, and these results should be confirmed by mRNA and Western blot analysis.”

4. Have authors determine the results in some cell lines or mouse model?

ANSWER: No, we did not performed experiments in cell lines and mouse models.

5. authors stated limitations of this study. Your future plan or perspective should be included.

ANSWER: We thank the reviewer for that suggestion. We added the following at the end of discussion:

Nevertheless, we provide the first evidence of the role of semaphorins in NAFLD. This was the first phase of the study aiming to depict the role of semaphorins in NAFLD and infections (project SepsisFAT). While we showed the association of serum concentrations with NAFLD, the question of their pathophysiological role in NAFLD remains unknown. The next step would be to show how they regulate T-cell immunological homeostasis and correlation with cytokine and growth factor networks in NAFLD patients, cell cultures and mice models.

6. the references in this manuscript were almost published 10 years ago

ANSWER: Some of the references were indeed published 10 years ago, which highlights the lack of data on the role of semaphorins in liver diseases, fibrogenesis and immune responses. There is more recent data on the role of semaphorins in oncogenesis, cancer biomarkers and in vitro experiments describing their oncogenic role, and more recently role in cardiovascular disease. We added 3 references in Introduction (FASEB J. 2022 Oct;36(10):e22509.; J Cell Physiol. 2021 Sep;236(9):6235-6248; Front Oncol. 2022 Jan 27;12:793805). However, we believe that describing in detail their role in oncogenesis in discussion section is out of the scope of our manuscript.

Round 2

Reviewer 2 Report

Authors revised the manuscript according to my comments. I suggest accepting it.